# Protective Effects of Sesamol against Liver Oxidative Stress and Inflammation in High-Fat Diet-Induced Hepatic Steatosis

**DOI:** 10.3390/nu13124484

**Published:** 2021-12-15

**Authors:** Wenya Zheng, Ziyu Song, Sha Li, Minmin Hu, Horia Shaukat, Hong Qin

**Affiliations:** 1Department of Nutrition Science and Food Hygiene, Xiangya School of Public Health, Central South University, 110 Xiangya Road, Changsha 410078, China; wenyazheng@csu.edu.cn (W.Z.); song0305ziyu@126.com (Z.S.); huminmin0229@126.com (M.H.); hooriyashaukat@gmail.com (H.S.); 2Changsha Center for Disease Control and Prevention, 509 Wanjiali North Road, Changsha 410005, China; ls7699459@163.com

**Keywords:** sesamol, hepatic steatosis, oxidative stress, inflammation

## Abstract

Chronic high-fat diet (HFD) is associated with the onset and progression of hepatic steatosis, and oxidative stress is highly involved in this process. The potential role of sesamol (SEM) against oxidative stress and inflammation at the transcriptional level in a mice model of hepatic steatosis is not known. In this study, we aimed to investigate the scavenging effects of SEM towards reactive oxygen generated by lipid accumulation in the liver of obese mice and to explore the mechanisms of protection. Markers of oxidative stress, vital enzymes involved in stimulating oxidative stress or inflammation, and nuclear transcription of Nrf2 were examined. Our results showed that SEM significantly inhibited the activity of the HFD-induced hepatic enzymes CYP2E1 and NOX2, associated with oxidative stress generation. Additionally, SEM reversed HFD-induced activation of NF-κB, a redox-sensitive transcription factor, and attenuated the expression of hepatic TNF-α, a proinflammatory molecule. Moreover, SEM enhanced HFD-induced hepatic Nrf2 nuclear transcription and increased the levels of its downstream target genes *Ho1* and *Nqo1*, which indicated antiinflammation and antioxidant properties. Our study suggests that chronic HFD led to hepatic steatosis, while SEM exhibited protective effects on the liver by counteracting the oxidative stress and inflammation induced by HFD. The underlying mechanism might involve multiple pathways at the transcriptional level; the antioxidant defense mechanism was in partly mediated by the upregulation of Nrf2.

## 1. Introduction

Non-alcoholic fatty liver disease (NAFLD) is considered a global epidemic. The population affected by NAFLD is expected to expand from 83.1 million in 2015 (approximately 25% of the population) to 100.9 million in 2030 [1]. It is one of the most common metabolic disorders that could be caused by abnormal or excessive lipids accumulation, oxidation, or transportation in the liver. The crucial pathobiological aspect of NAFLD is hepatic steatosis that has the tendency to progress to non-alcoholic steatohepatitis (NASH) or even cirrhosis and liver cancer [2,3]. It is well known that several exogenous factors such as a chronic high-fat diet (HFD) or other unhealthy lifestyle habits cause obesity and in parallel influence the onset and progression of hepatic steatosis [4]. During the process from health liver to simple steatosis and further to severe liver disease, chronic oxidative stress and inflammation are play important roles [5]. Therefore, antioxidants and anti-inflammatory factors are crucial for treating hepatic steatosis and slowing the progress of NAFLD.

Oxidative stress is the result of an imbalance between the activity of oxidant and antioxidant enzymes, which leads to the production of an excess of reactive oxygen species (ROS) [6]. Many studies have demonstrated that an excessive ROS generation and accumulation is involved in the progress of NAFLD [7,8]. Regarding the source of ROS, they are mainly generated in mitochondria and the endoplasmic reticulum. Mitochondria are the main organelles that produce ROS by inducing the uncoupling of the electron transport. Oxidative stress in hepatic steatosis may accelerate the progression of the disease, in which enzymes like NADPH oxidases and oxidoreductase cytochrome are involved. The regulation of the activity of multiple enzyme systems might alleviate HFD-induced steatotic liver. Moreover, a long-term oxidative stress is strongly correlated with increased inflammation [9]. The nuclear factor erythroid 2-related factor 2 (Nrf2) is a major regulator of the antioxidant defense system and of inflammation [10], acting as an important etiological factor in hepatic steatosis progression.

In patients with progressive NAFLD, the production of antioxidants is repressed, as shown by both histopathological and molecular biological analyses [11,12]. The total antioxidant activity is insufficient to compensate for oxidative stress. Therefore, it is speculated that agents that promote cellular antioxidant defense systems have therapeutic potential and may be useful to prevent or treat NAFLD [13]. Recently, an increasing number of studies have focused on the use of phytochemicals as natural antioxidants. Sesamol (SEM), a major constituent of sesame seed oil, has been reported to improve the cognitive impairment caused by oxidative stress, and its protective effect on neurons is partly related to the activation of Nrf2 [14]. Our previous study demonstrated that SEM is able to attenuate hepatic steatosis in HFD-induced obese mice via a PKA pathway [15]. However, the role of SEM in regulating hepatic redox homeostasis, inflammation, and hepatic steatosis caused by HFD is not yet clear. Therefore, in this study we aimed to investigate the effects and potential molecular transcription targets of SEM in alleviating oxidative stress and inflammation in the liver of obese mice induced by HFD and to illustrate the beneficial effects of SEM in treating and preventing NAFLD.

## 2. Materials and Methods

### 2.1. Animals and Diet

Eight-week-old male C57BL/6J mice were brought from Central South University (Hunan, Changsha, China). All animals were kept under controlled conditions of illumination with a 12:12 h light–dark cycle and constant room temperature (22 ± 2 °C, humidity 40–70%). The mice were allowed to free feed and drinking water, with one week of adaptive feeding. The mice were divided into three groups: (1) normal-fat diet (CON) group, (2) HFD group, (3) HFD + SEM group. All animal experiments were performed in accordance with the protocol (Approval Number: XYGW-2019-038) approved by the Institutional Animal Care and Use Committee of Central South University.

The mice in the CON group were fed a diet consisting of 10 kcal% fat, 70 kcal% carbohydrate, and 20 kcal% protein (9D12450J, Research Diets Inc., New Brunswick, NJ, USA). The rest of the mice were fed an HFD consisting of 60 kcal% fat, 20 kcal% carbohydrate, and 20 kcal% protein (D12492, Research Diets Inc., New Brunswick, NJ, USA) for 8 weeks according to the established model of obese mice. The mice whose weight was 20% higher than the average weight of the mice in the CON group were considered obese and were randomly divided into two groups. These two groups were fed continuously the HFD for another 8 weeks, and one group also received SEM. SEM was dissolved in a vehicle (0.5% carboxymethyl cellulose) and gavaged into mice of the HFD + SEM group at a dose of 100 mg/kg body weight once daily, while the other mice were given an equal volume of vehicle, also by gavage. The body weight of all mice was measured once a week, and food intake levels were recorded once a day.

### 2.2. Serum and Tissue Collection

After 16 weeks, blood samples were collected from the femoral artery and stored overnight at 4 °C. Then, serum was isolated by centrifuging the samples at 3000× rpm for 15 min. These samples were stored at −80 °C. The mice were sacrificed when 25 weeks old; their livers were dissected, rinsed in normal saline, and weighed. The livers were further used for expression analysis of proteins and mRNAs. 

### 2.3. Serum Parameter Analysis

Malondialdehyde (MDA) and superoxide dismutase (SOD) were measured by commercial kits bought from Jiancheng Bioengineering Institute (Nanjing, China). All steps were carried out according to the manufacturer’s instructions protocol.

### 2.4. Hepatic Parameters Analysis

The liver tissue was homogenized in normal saline by centrifugation, and the supernatant was collected. SOD and MDA levels in the liver samples were measured by commercial kits bought from Jiancheng Bioengineering Institute (Nanjing, China).

### 2.5. Western Blotting Analysis

One part of the liver tissue was quickly frozen in liquid nitrogen and immediately stored at −80 °C until use. The liver tissues from at least 8 animals of each group were harvested in cold RIPA buffer containing a proteinase inhibitor (Ding Guo, Beijing, China) and then centrifuged at 15,000× *g* for 40 min at 4 °C. The protein level was determined using a BCA assay kit for the following western blot analysis. Protein samples were separated by sodium dodecyl sulfate–polyacrylamide gel electrophoresis and blotted onto a PVDF membrane. The membranes were blocked in 5% nonfat dry milk or 3% bull serum albumin, and then primary antibodies were added at 4 °C overnight. Proteins were detected using the primary antibodies anti-cytochrome P450 2E1 (CYP2E1, 1:1000), anti-NADPH-oxidase 2 (NOX2, 1:1000), anti-uncoupling protein 2 (UCP2, 1:1000), anti-NF-κB (1:1000), anti-TNF-α (1:1000), anti-Nrf2 (1:1000) and anti-β-actin (1:300,000). The antibodies above were purchased from Abclonal (Boston, MA. USA). Then, the membranes were incubated with secondary antibodies for 1 h at room temperature. The protein bands were detected using a chemiluminescence imager (Tanon-5500, Shanghai, China) after ECL, and band densities were quantified via densitometry by Image J.

### 2.6. Quantitative Reverse-Transcription-Polymerase Chain Reaction (RT-PCR)

RNA samples from at least 8 animals of each group were prepared from 50 mg of liver homogenates using Trizol reagent. cDNA synthesis was performed for each group with the same amount of RNA from each mouse, using a kit purchased from Vazyme (Nanjing, China). The levels of gene expression were determined using a SYBR Green Master Mix (Yeasen Biotech, Shanghai, China) and analyzed on a LightCycler 480 II (Roche, Basel, Switzerland). The forward and reverse primers we used are shown in Table 1.

### 2.7. Statistical Analysis

The data are presented as min to max in box-and-whisker diagrams, and statistical analysis was performed using SPSS 20.0 (Chicago, IL, USA). One-way ANOVA was used when the data involved three or more groups. The LSD test was used to compare the results between two groups for statistical significance. For all statistical tests and graphical presentations, the Prism 8.0 software was used. The results were considered statistically significant at *p* < 0.05.

## 3. Results

### 3.1. Body Weight and Energy Intake

Mice in all three groups had the same average body weight at the beginning of the experiments. After supplying the HFD or HFD + SEM, the weight of the mice fed the HFD significantly increased compared to that of the mice fed a normal-fat diet, while the 8-week treatment with SEM significantly decreased the weight of the HFD mice (Figure 1A). However, there was no significant difference in energy intake between mice administered HFD and HFD + SEM (Figure 1B). Similarly, liver weight and body fat were remarkably increased by the HFD, whereas the values of these two parameters were significantly lower in mice treated with SEM (Figure 1C,D). Additionally, as shown in Figure 1E,F, the liver weight was positively correlated with body weight or body fat (*p* < 0.05), indicating a correlation between hepatic steatosis and obesity.

### 3.2. Markers of Oxidative Stress in Serum

As shown in Figure 2A, the levels of MDA in serum samples were significantly increased in the HFD group compared with the CON group. Interestingly, the administration of SEM markedly reduced the MDA serum levels, which indicated that the HFD-induced systematic oxidative stress was improved after SEM treatment. In addition, liver weight was positively correlated with serum MDA levels (Figure 2C), suggesting serum MDA was associated with hepatic steatosis. However, serum SOD levels were not significantly changed by the interventions (Figure 2B).

### 3.3. Effects of SEM on Hepatic MDA and SOD Production

The levels of hepatic MDA were dramatically induced by HFD (six-fold), while the content of MDA was dramatically inhibited by SEM treatment in HFD mice (Figure 3A). Moreover, the levels of hepatic SOD were significantly suppressed by HFD (0.5-fold), whereas gavage of SEM in HFD-fed mice significantly restored the SOD levels (Figure 3B). Additionally, as expected, liver weight was correlated with hepatic MDA as well as SOD concentrations (Figure 3C,D).

### 3.4. Effects of SEM on Mediators Involved in Oxidative Stress Generation in Fatty Liver

To verify if SEM may protect mice from hepatic steatosis through its antioxidant activity, we analyzed the expression of the enzymes CYP2E1 and NOX2 as well as of the uncoupling protein UCP2, which are all importantly involved in oxidative stress generation. HFD feeding resulted in a significant increase in protein expression of CYP2E1 (Figure 4A), NOX2 (Figure 4B), and UCP2 (Figure 4C) in the liver, while SEM treatment significantly reduced the activity of the three enzymes. Further, the mRNA levels of *cyp2e1* (Figure 4D), *nox2* (Figure 4E), and *ucp2* (Figure 4F) were similarly modulated as their corresponding protein levels. The mRNAs encoding CYP2E1, NOX2, and UCP2 proteins were also suppressed by SEM in the liver of HFD mice.

### 3.5. Effects of SEM on Inflammatory Factors in Fatty Liver

HFD-induced obesity resulted in the increased secretion of several inflammatory cytokines such as TNF-α and IL-1β. NF-κB controls the global proinflammatory response; therefore, the hepatic protein and mRNA expression of NF-κB and TNF-α, two important proinflammatory cytokines, was investigated in this study. As we speculated, HFD significantly induced hepatic NF-κB and TNF-α levels up to 1.4 and 2-fold compared to the CON group, respectively (Figure 5A,B). SEM also showed beneficial effects in reducing NF-κB and TNF-α levels back to normal values. The mRNA expression of *nf-κb* and *tnf-α* was regulated in a similar way (Figure 5C,D).

### 3.6. Effects of SEM on Hepatic Regulators of Oxidative Stress and Inflammation

Nrf2, a transcription factor, is a well-known defense protein against oxidative stress in various tissue injury models. To further investigate whether Nrf2 was activated in SEM-induced protection from obesity-associated hepatic steatosis, the expression of Nrf2 and its downstream targets heme oxygenase 1 (Ho1) and recombinant NADH dehydrogenase quinone 1 (Nqo1) was examined. As shown in Figure 6A, Nrf2 protein expression remained unchanged in the HFD group compared to the CON group. However, the administration of SEM significantly induced Nrf2 protein levels in HFD mice, indicating the activation of Nrf2 by SEM in the presence of hepatic steatosis. Similarly, the mRNA levels of Ho1 (Figure 6B) and Nqo1 (Figure 6C) were significantly upregulated by SEM administration in our HFD-fed animals.

## 4. Discussion

Oxidative stress and inflammation play essential roles in the pathological progression of hepatic steatosis, which might develop into NAFLD, NASH, or cirrhosis [2]. In the current study, we explored the novel therapeutic effects and the potential targets of SEM in relieving oxidative stress and inflammation related to hepatic steatosis in the HFD-induced obese C57BL/6J mice model. Our key findings are that SEM exhibited a protective role on liver health by reducing the generation of hepatic oxidative stress and inflammation. These effects could be mediated through the activation of the transcription factor Nrf2.

Hepatic steatosis is common in obese individuals and is associated with multiple metabolic disorders such as insulin resistance, hyperlipidemia, and atherosclerosis [16]. In the current study, HFD-fed mice showed increased body weight, body fat, and liver weight, which led to hepatic steatosis, as illustrated and proved in our previous study by histological analysis of the liver [15]. Our results showed that hepatic steatosis was ameliorated by SEM treatment (100 mg/kg body weight once daily) without affecting the energy intake from the diet. We chose the dosage of SEM in our experiments by referring to several other relevant metabolic studies which also gavaged mice with 100 mg/kg body weight of SEM [17,18,19]. In addition, our previous study found that the administration of SEM at 100 mg/kg body weight had the capacity to prevent obesity and related hepatic steatosis by regulating genes or proteins involved in lipogenesis, lipolysis, and fatty acid β-oxidation [15,20]. In our current study, we further demonstrated that SEM modulated markers such as MDA and SOD, which are associated with oxidative stress in mice models of hepatic steatosis. MDA is one of the low-molecular-weight end products of fatty acid peroxidation and it is a good indicator of the degree of oxidative stress. The products of lipid peroxidation can be eliminated by enzymatic antioxidant defense systems. SOD is part of the enzymatic antioxidant defense systems, which can help cells eliminate the toxic effects induced by free radicals. SOD is also a physiological antioxidant that can prevent subsequent lipid peroxidation [21]. In the presence of hepatic steatosis, an imbalance between MDA generation and SOD activity was observed [22,23]. These findings are in agreement with our results. Moreover, some studies established that SEM attenuated oxidative damage in some pathological conditions [14,24], and these effects were also shown in our current study. However, those studies were mainly focused on the protective impact of SEM on oxidative injury induced by neurodegeneration or sepsis. Our group showed the beneficial impact of SEM in reducing hepatic oxidative stress as well as inflammation in a mice model of HFD-induced hepatic steatosis, which indicated that SEM might stop NAFLD progression, since oxidative stress and inflammation are normally considered as a “second hit” after a “first hit” in the process.

CYP2E1, a member of the oxidoreductase cytochrome family mainly present in the mitochondrial inner membrane and the endoplasmic reticulum, is responsible for oxidizing a variety of small molecule substrates including fatty acids [22]. Many studies showed that the capacity of CYP2E1 to generate ROS significantly contributes to oxidative stress in NAFLD [25]. The reason could be that CYP2E1 acts as an important transcription factor, increasing MDA and decreasing SOD levels [22]. The absence or block of CYP2E1 may exert beneficial effects on NAFLD by inhibiting oxidative stress [22,26]. Our data suggest that HFD induced the overexpression of both mRNA and protein levels of CYP2E1 in hepatic steatosis, while SEM significantly reverted the transcription of *cyp2e1*, thus reducing the oxidative stress.

The CYP pathway is known to be coupled with the NOX pathway [27]. Endogenous production of O_2_ arises from the NOX family. Several NOX isoforms play important roles in sustaining the progression of various chronic liver diseases, e.g., NAFLD and NASH, by generating oxidative stress. It has been observed that a diet rich in fructose can upregulate the expression of both NOX2 and NOX4 in the liver. The increased expression of CYP2E1 leads to the generation of more free electrons, a process coupled with the conversion of NADPH to NADP+ via NOX2 and/or NOX4. In particular, NOX2 is responsible for phagocytic activity and inflammation [28]. Our results showed that HFD induced the overexpression of NOX2 at both mRNA and protein levels, and these effects were markedly reversed by treatment with SEM, which is in line with the expression pattern of CYP2E1 in our current results.

Previous studies indicated that the upregulation of UCP2 could slow the process of oxidative phosphorylation and inhibit ATP production, thereby causing energy metabolism disorders in cells [29]. UCP2 is found in the mitochondria and functions by controlling ROS in lipid metabolism. Actually, the physiological function of UCP2 is still controversial and, for example, one study showed protective effects of UCP2 induction after liver ischemia–reperfusion injury in obese mice [30]. However, several studies using the same animal model as in our study pointed out increases in UCP2 expression in obese subjects or in animals with a metabolic disorder in response to a high-fat diet [31,32]. This might be due to mitochondrial saturation of free fatty acids, and therefore, an increase in the production of mitochondrial peroxidation, generating a greater quantity than required of ROS, which stimulates the expression of UCP2, as lipid hydroperoxides trigger its production [31]. In our study, UCP2 acted as a negative regulator of the mitochondrial function after hepatic steatosis in obese mice. The effect of SEM on hepatic UCP2 expression was also investigated. As expected, SEM markedly restrained the increasing hepatic UCP2 expression observed in HFD-fed mice. Therefore, these results revealed the therapeutic effect of SEM on oxidative stress in mice with HFD-induced hepatic steatosis, indicating that SEM is a natural product that might alleviate HFD-induced NAFLD-associated oxidative stress via inhibiting CYP2E1, NOX2, and UCP2.

Inflammation might occur when the production of ROS is sustained and antioxidant enzymes cannot balance it [6]. In the current study, we found that the protein and mRNA expression of the proinflammatory cytokines NF-κB and TNF-α was significantly downregulated by SEM administration in HFD mice that exhibited very high expression of inflammatory factors in the liver, which is in accordance with the results of several other studies [15,20]. Nrf2, a redox-sensitive transcription factor, is a crucial functional molecule involved in ROS scavenging by activating the transcription of various of antioxidants such as SOD [33]. As shown by our results, the activity of SOD was clearly modulated by SEM, indicating that Nrf2 might be involved in the antioxidative and anti-inflammatory effects induced by SEM. In a resting cell, Nrf2 mainly resides in the cytosol and is bound to its negative regulator Kelch-like ECH-associated protein 1 (Keap1). The coupling between Nrf2 and Keap1 leads to Nrf2 proteasomal degradation [34,35]. When activated by a stimulus, Nrf2 and Keap1 are separated, which leads to the elevation of free Nrf2 levels and its translocation to the nucleus [14,34]. The activation of Nrf2 and the transcription of its target genes, including Ho1 and Nqo1, commonly analyzed together to illustrate the activation of Nrf2, are important for hepatocyte antioxidant defense [36]. In our study, we observed that SEM significantly enhanced the protein expression of Nrf2; additionally, the expression of its target genes Ho1 and Nqo1 was upregulated, which suggested transcriptionally activation of Nrf2 by SEM in the liver of obese mice. Interestingly, HFD also slightly increased the activation of Nrf2 transcription, which might due to a feedback of oxidative stress stimuli. However, SEM treatment further transcriptionally activated Nrf2 compared to the levels observed in the liver of HFD mice. The modulation of Nrf2 transcription in oxidative injury by the administration of antioxidant substances has been reported in other previous studies [36,37] and is in accordance with our results.

The Nrf2 signaling pathway is related to inflammation. The absence of the Nrf2 antioxidant signaling exacerbates the inflammatory response mediated by NF-κB and inflammatory cytokines such as TNF-α and IL-1β, finally leading to liver injury [38]. Moreover, some research reported NF-κB-mediated transcriptional regulation of rat CYP2E1 in alcohol-induced liver injury [39], which was partly proved by our results. Therefore, Nrf2 is highly involved in generating oxidative stress and inflammation. Since our results showed that Nrf2 was parallelly modulated with markers of oxidative stress and inflammation, it is suggested that Nrf2 mediated the regulation of oxidative stress and inflammation following SEM treatment under the condition of hepatic steatosis.

## 5. Conclusions

The present study provides new evidence that SEM alleviated HFD-induced hepatic steatosis by reducing oxidative stress and inflammation in the liver. The positive effects mediated by SEM might be through the activation of Nrf2 transcription (Figure 7). Together, these findings make an essential contribution to understanding the biological effects of SEM. They provide a theoretical basis and scientific evidence for supporting the use of SEM, as a bioactive antioxidant compound, to treat high-fat diet-induced hepatic steatosis. Moreover, since techniques of extracting it from sesame seeds are well established, SEM could be expected to be developed into a new functional compound or even a potential therapeutic agent extracted from common food. In the future, human intervention studies or clinical trials may support the utilization of SEM to prevent the progression of NAFLD.

## Figures and Tables

**Figure 1 nutrients-13-04484-f001:**
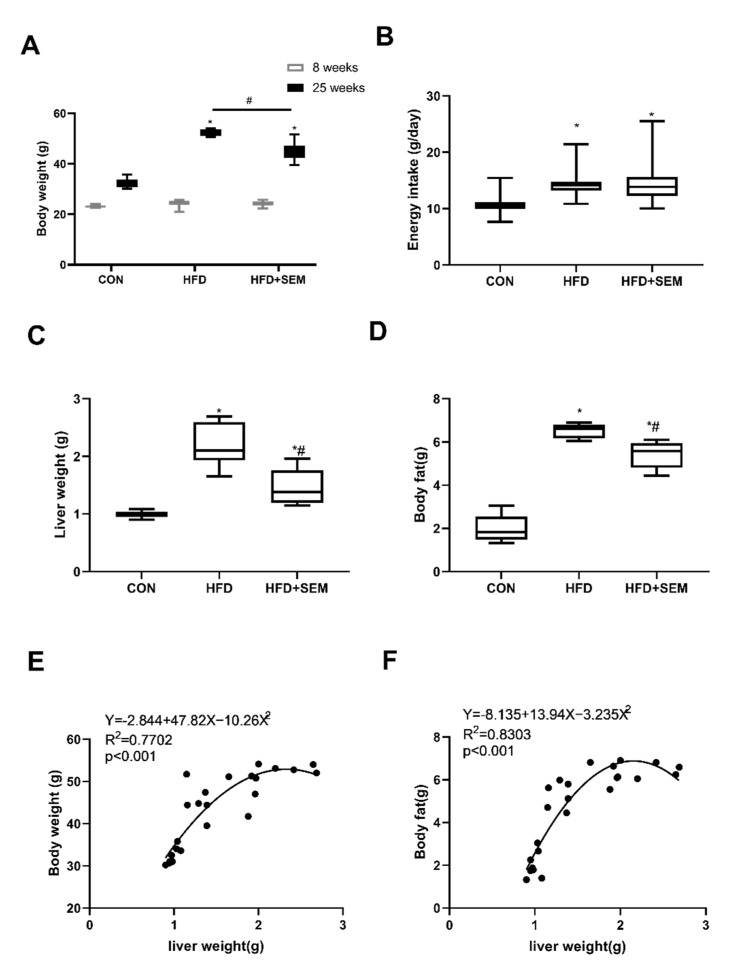
Sesamol (SEM) improves hepatic steatosis in high-fat diet (HFD)-fed mice. Effects of HFD and HFD + SEM on (**A**) body weight, (**B**) energy intake, (**C**) liver weight, (**D**) body fat, (**E**) correlation of body weight with liver weight, and (**F**) of body fat versus with weight. Box-and-whisker diagrams are presented as min to max and analyzed with one-way ANOVA; * marks significant differences compared to the control group (*p* ≤ 0.05); # marks significant differences between the HFD group and the HFD + SEM group (*p* ≤ 0.05). Correlation data (**E**,**F**) were fitted to a second-order polynomial. Equation, r^2^ and *p*-value are shown in the left upper corner. Statistical analysis was performed by the Spearman’s rank correlation test; *p* ≤ 0.05 shows a statistically significant relation.

**Figure 2 nutrients-13-04484-f002:**
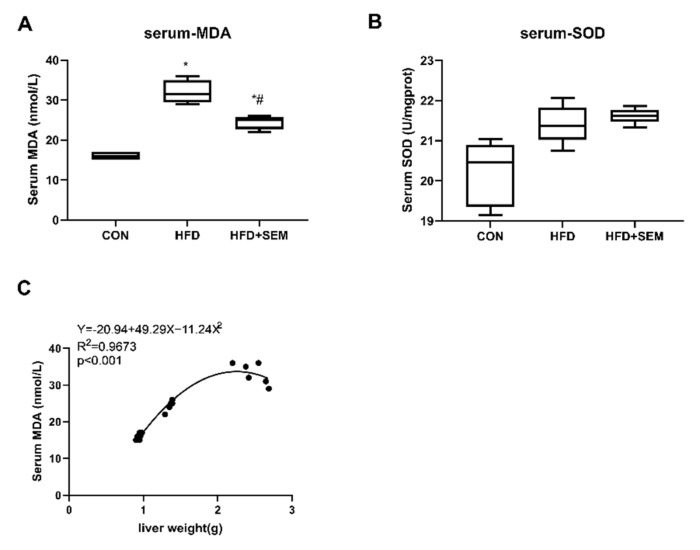
SEM treatment ameliorates hepatic steatosis-associated systemic oxidative stress. Effects of HFD and HFD + SEM on (**A**) serum MDA levels, (**B**) serum SOD levels, and (**C**) correlation of serum MDA levels with liver weight. Box-and-whisker diagrams are presented as min to max and were analyzed with one-way ANOVA; * vs. control group (*p* ≤ 0.05); # indicates significantly different when comparing mice fed HFD alone vs. mice fed HFD + SEM (*p* ≤ 0.05). Correlation data (**C**) are fitted to a second-order polynomial. Equation, r^2^ and *p*-value are shown in the left upper corner. Statistical analysis was performed by the Spearman’s rank correlation test; *p* ≤ 0.05 shows a statistically significant relation.

**Figure 3 nutrients-13-04484-f003:**
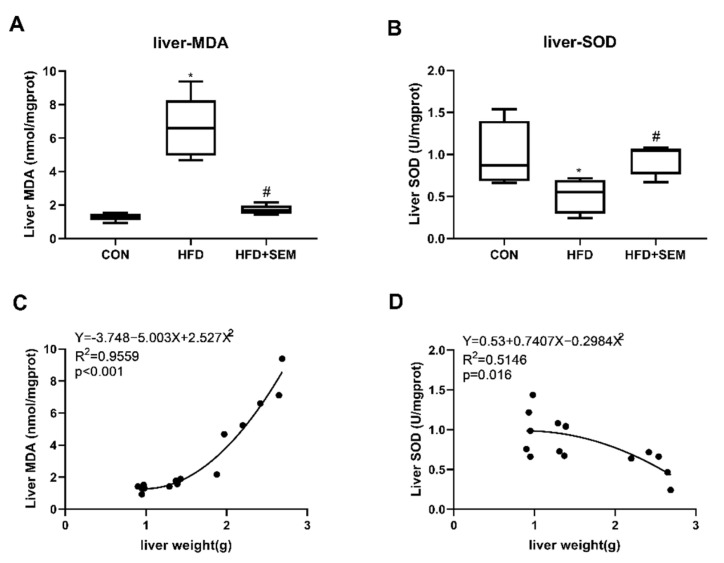
SEM treatment ameliorates oxidative stress in the liver of mice with hepatic steatosis caused by HFD. Effects of HFD and HFD + SEM on (**A**) MDA secretion in the liver, (**B**) SOD secretion in the liver, (**C**) correlation of hepatic MDA secretion with liver weight, and (**D**) correlation of hepatic SOD secretion with liver weight. Box-and-whisker diagrams are presented as min to max and were analyzed with one-way ANOVA; * vs. control group (*p* ≤ 0.05); # vs. HFD-alone treatment group (*p* ≤ 0.05). Correlation data (**C**,**D**) are fitted to a second-order polynomial. Equation, r^2^ and *p*-value are shown in the left upper corner. Statistical analysis was performed by the Spearman’s rank correlation test; *p* ≤ 0.05 shows a statistically significant relation.

**Figure 4 nutrients-13-04484-f004:**
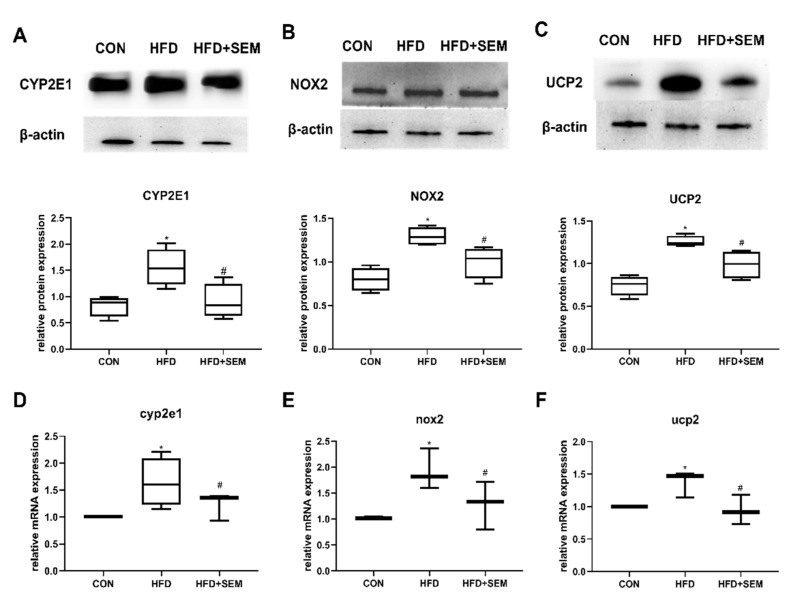
SEM treatment prevents HFD-induced hepatic steatosis by inhibiting protein and mRNA expression of enzymes involved in oxidative stress generation in the liver. Effects of HFD and HFD + SEM on the hepatic protein expression of (**A**) CYP2E1, (**B**) NOX2, (**C**) UCP2, and on hepatic relative mRNA expression of (**D**) *cyp2e1*, (**E**) *nox2*, (**F**) *ucp2*. The analysis of the relative intensities quantified by densitometry on western blots is presented in the lower panels. Data are presented as min to max in box-and-whisker diagrams and were analyzed with one-way ANOVA; * marks significant differences compared to the control group (*p* ≤ 0.05); # marks significant differences between the HFD group and the HFD + SEM group (*p* ≤ 0.05).

**Figure 5 nutrients-13-04484-f005:**
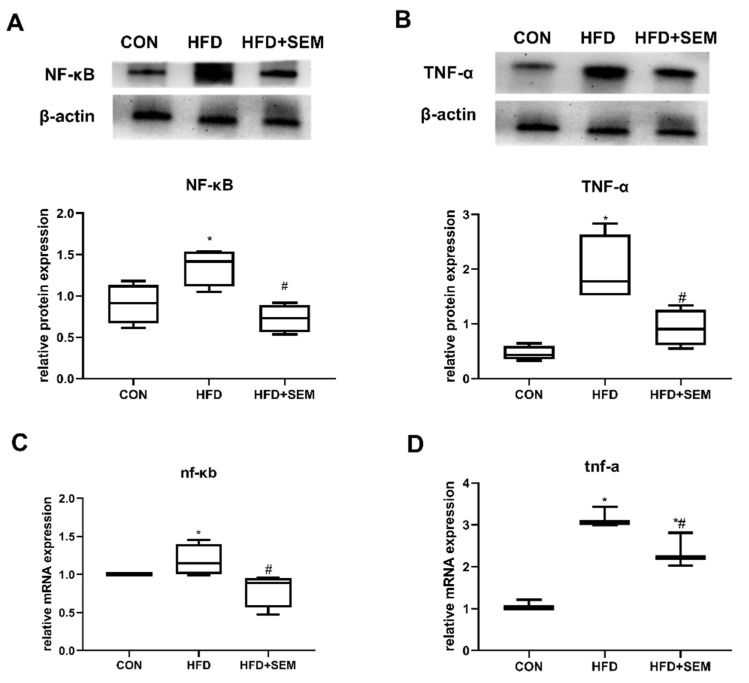
SEM treatment attenuated liver inflammation caused by HFD-induced hepatic steatosis. Effects of HFD and HFD + SEM on the hepatic expression of the proinflammatory proteins (**A**) NF-κB and (**B**) TNF-α and on the expression of the proinflammatory mRNAs of (**C**) *nf-κb* and (**D**) *tnf-α*. The relative intensities quantified by densitometry on western blots is presented in the lower panels. Data are presented as min to max in box-and-whisker diagrams and were analyzed with one-way ANOVA; * marks significant differences compared to the control group (*p* ≤ 0.05); # marks significant differences between the HFD group and the HFD + SEM group (*p* ≤ 0.05).

**Figure 6 nutrients-13-04484-f006:**
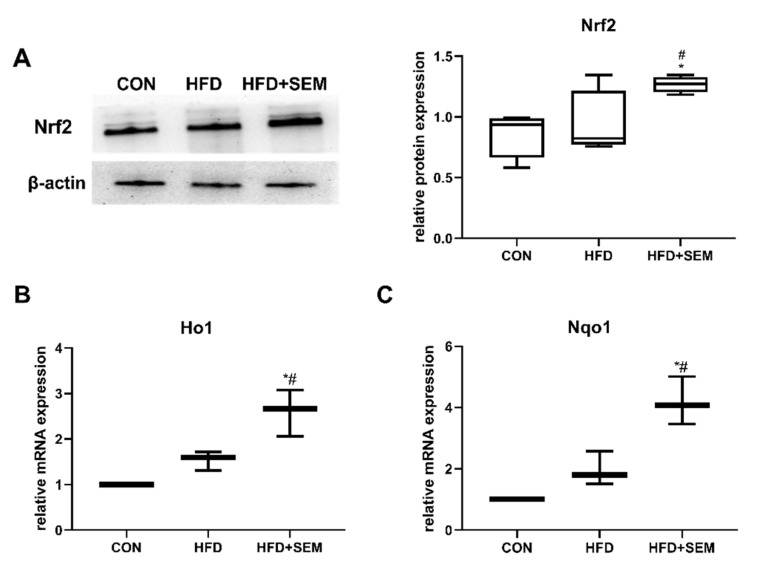
SEM prevents HFD-induced hepatic steatosis by modulating Nrf2 and its target genes Ho1 and Nqo1. Effects of HFD and HFD + SEM on hepatic (**A**) protein expression of Nrf2 and on relative mRNA expression of (**B**) Ho1 and (**C**) Nqo1. The relative intensities quantified by densitometry on western blots is presented in the panel on the right of the corresponding image. Data are presented as min to max in box-and-whisker diagrams and were analyzed with one-way ANOVA; * marks significant differences compared to the control group (*p* ≤ 0.05); # marks significant differences between the HFD group and the HFD + SEM group (*p* ≤ 0.05).

**Figure 7 nutrients-13-04484-f007:**
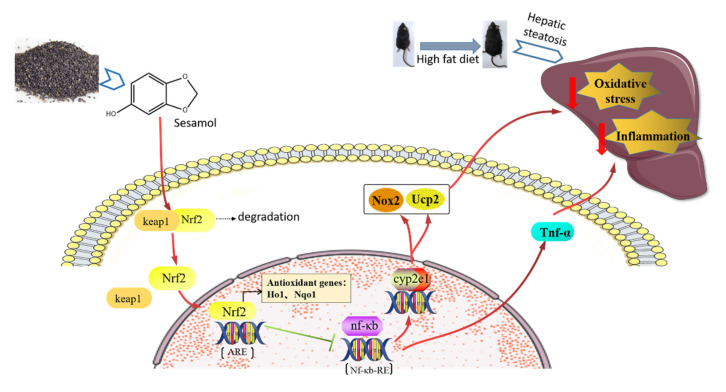
SEM alleviated HFD-induced hepatic steatosis by reducing oxidative stress and inflammation in the liver via activating Nrf2 transcription.

**Table 1 nutrients-13-04484-t001:** Primer Sequences.

Name	Primer Sequence (5′ to 3′)
cyp2e1	Forward: GTGACTGGGGAATGGGGAAA
	Reserve: AGGTAGGGTCAAAAGGCTGG
nox2	Forward: TGTGAGAGGTTGGTTCGG
	Reserve: CAGGAGCAGAGGTCAGTGTG
ucp2	Forward: TTCTCCCAATGTTGCCCG
	Reserve: CCCGAAGGCAGAAGTGAAGT
nf-κb	Forward: GACCTGGTTTCGCTCTTG
	Reserve: TGCTGTATCCGGGTACTT
tnf-α	Forward: CACCACGCTCTTCTGTCTACTGAAC
	Reserve: AGATGATCTGAGTGTGAGGGTCTGG
Ho1	Forward: ACCGCCTTCCTGCTCAACATTG
	Reserve: CTCTGACGAAGTGACGCCATCTG
Nqo1	Forward: GCGAGAAGAGCCCTGATTGTACTG
	Reserve: AGCCTCTACAGCAGCCTCCTTC
β-actin	Forward: CGTGCGTGACATCAAAGA
	Reserve: AAGGAAGGCTGGAAAAGA

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
