# Peer review of "Protective Effects of Sesamol against Liver Oxidative Stress and Inflammation in High-Fat Diet-Induced Hepatic Steatosis"

_nutrients, 2021, doi:10.3390/nu13124484_

Round 1

Reviewer 1 Report

In this paper by Zheng et al, the Authors investigated the ameliorative effect of sesamol supplementation in rats fed with a high fat diet. In particular, the Authors have shown as sesamol reduced body and liver weight, and MDA liver concentration respect the HFD group. At the same time, the sesamol counteracted the effect of HFD on oxidative stress markers via NRF2 induction. The study is quite elegant, well written, the methodologies used are pertinent and the results give new insight on the beneficial role of sesamol to ameliorate the physio-phatological condition on NAFLD. Despite these considerations, some issue arises:

Major consideration

  1. Considering that the average sesamol concentration founded in sesame seeds and oil is 6 and 60 mg sesamol/kg respectively (doi.org/10.3390/foods9040394), it means that people with an average body weight of 70kg should introduce from 1,1kg to 120g of sesame seeds and sesame oil respectively to reach the same experimental condition in rats (100mg/kg body weight). Of course this is not realizable in a normal contest. The Authors should underline the real exploitation of the condition selected in the study

Minor consideration

  1. If the Authors state that SOD gene is a target of NRF2 (line 332-334), why they did not investigate its transcriptional expression as Ho1 and Nqo1?

Reviewer 2 Report

First of all, I appreciate the trust placed in me to review the present work.
I think it has been done very correctly and is written with great clarity, the results being consistent with the objectives set.
I also think that it can be published without any modification, only the change in the graphs of the bar model for that of box diagrams could give greater clarity to those results.

Round 2

Reviewer 1 Report

i have no further comments